# Bisphenol A and Type 2 Diabetes Mellitus: A Review of Epidemiologic, Functional, and Early Life Factors

**DOI:** 10.3390/ijerph18020716

**Published:** 2021-01-15

**Authors:** Francesca Farrugia, Alexia Aquilina, Josanne Vassallo, Nikolai Paul Pace

**Affiliations:** 1Department of Physiology and Biochemistry, University of Malta, MSD 2080 Msida, Malta; francesca.farrugia.18@um.edu.mt (F.F.); alexia.b.aquilina.18@um.edu.mt (A.A.); josanne.vassallo@um.edu.mt (J.V.); 2Centre for Molecular Medicine and Biobanking, University of Malta, MSD 2080 Msida, Malt

**Keywords:** bisphenol A, type 2 diabetes, beta cell, endocrine disruptors

## Abstract

Type 2 diabetes mellitus (T2DM) is characterised by insulin resistance and eventual pancreatic β-cell dysfunction, resulting in persistent high blood glucose levels. Endocrine disrupting chemicals (EDCs) such as bisphenol A (BPA) are currently under scrutiny as they are implicated in the development of metabolic diseases, including T2DM. BPA is a pervasive EDC, being the main constituent of polycarbonate plastics. It can enter the human body by ingestion, through the skin, and cross from mother to offspring via the placenta or breast milk. BPA is a xenoestrogen that alters various aspects of beta cell metabolism via the modulation of oestrogen receptor signalling. In vivo and in vitro models reveal that varying concentrations of BPA disrupt glucose homeostasis and pancreatic β-cell function by altering gene expression and mitochondrial morphology. BPA also plays a role in the development of insulin resistance and has been linked to long-term adverse metabolic effects following foetal and perinatal exposure. Several epidemiological studies reveal a significant association between BPA and the development of insulin resistance and impaired glucose homeostasis, although conflicting findings driven by multiple confounding factors have been reported. In this review, the main findings of epidemiological and functional studies are summarised and compared, and their respective strengths and limitations are discussed. Further research is essential for understanding the exact mechanism of BPA action in various tissues and the extent of its effects on humans at environmentally relevant doses.

## 1. Introduction

Type 2 diabetes mellitus (T2DM) is a common chronic metabolic disorder characterised by peripheral insulin resistance, β-cell dysfunction and an inadequate compensatory insulin secretory response [1]. Persistent hyperglycaemia leads to the development of both microvascular and macrovascular complications, including retinopathy, neuropathy, nephropathy, and an increased incidence of atherosclerotic disease [2,3]. The disorder has an underlying complex aetiology, with several established risk factors including sedentary lifestyle, calorie dense diets, visceral adiposity and a broad array of both common and rare genetic variants [4,5,6]. In addition to these classical risk factors, an increasing body of evidence implicates environmental chemicals in the rising epidemic of T2DM. Endocrine disrupting chemicals (EDCs) are exogenous chemicals that have adverse metabolic consequences as they interfere with the synthesis, secretion, transport, binding, action and metabolism of endogenous hormones [7,8]. Bisphenol A (BPA) is a widespread EDC that is the main constituent of polycarbonate plastics. It is used in the manufacture of epoxy resins, in the lining of food cans, as well as in recycled paper, carbonless cash register receipts and the coating of CDs and DVDs [9,10]. BPA from these products reacts with chlorinated tap water to form chlorinated BPA derivatives [11]. The primary route of human exposure to BPA is oral, and over 90% of individuals have detectable urine BPA levels even though it is a non-persistent EDC with a short half-life [12,13]. Human pharmacokinetic studies indicate that following single exposure by ingestion, BPA undergoes rapid hepatic conjugation and excretion through bile and urine with a half-life of approximately 5.3 h [14].

BPA is a structural analogue of endogenous 17β-oestradiol and its precise mechanisms of action at the cellular level are not fully understood. As a xenoestrogen, it binds and acts via extranuclear oestrogen receptors (ERα and ERβ) at environmentally relevant doses [15,16]. ERs are expressed in a wide variety of tissues, including ovaries, testes, prostate, liver, breast, brain, bone marrow, adipose tissue and the pancreas [17]. BPA can act in a cell-type specific manner as an ER agonist at concentrations greater than 10 nM, or as an ER antagonist at concentrations less than 10 nM in vitro [18,19]. BPA additionally acts through G protein-coupled receptor 30 (GPR30)—a 7-transmembrane G-protein coupled receptor that mediates the rapid non-genomic signal transduction of oestrogens [20,21]. BPA has relatively high affinity for GPR30, through which it induces the rapid activation of the ERK signalling pathways [14,22]. It has also been shown to inhibit the release of adiponectin, and to exert proangiogenic effects on endothelium [23,24]. Adiponectin has insulin-sensitising, anti-atherogenic and anti-inflammatory properties, and hypoadiponectinemia has been associated with insulin resistance and T2DM [25,26,27,28]. BPA also increases the expression of several proinflammatory adipocytokines, including interleukin-6 (Il-6) and monocyte chemoattractant protein 1α (MCP1α) through GPR30 [29]. Despite the potential mechanistic pathways linking BPA exposure to metabolic disease, several population-based epidemiological studies have shown contradictory or conflicting associations between BPA and T2DM risk. The aim of this article was to (1) provide an updated review of the salient clinical and population studies on BPA and T2DM; (2) review the core mechanistic evidence from functional studies that provides physiologic plausibility to the observed epidemiological associations; (3) relate the association between BPA and T2DM to early life exposure; and (4) outline the emerging mechanisms that could explain the pleiotropic effects of BPA on multiple body systems.

## 2. Materials and Methods 

This systematic literature review search is reported in accordance with the Preferred Reporting Items for Systematic Reviews and Meta-Analyses statement guidelines [30]. The article selection process is summarised in a Preferred Reporting Items for Systematic Reviews and Meta-Analyses (PRISMA) flow diagram (Appendix A).

### 2.1. Study Selection and Inclusion Criteria

To identify all studies that examine the relationship between BPA and T2DM, we conducted an electronic literature search in PubMed and Medline databases. The search covered articles published between January 2006 and July 2020 and was restricted to articles published in the English language. The following combinations of free text keywords and Medical Subject Heading terms were used in the search: ‘Diabetes Mellitus, type 2′ (MeSH) AND ‘bisphenol A’ OR ‘BPA’. Epidemiological studies that provided population-level measures of association included observational, case-control, cross-sectional studies and meta-analysis. Functional in vivo or in vitro studies that aimed to investigate the mechanistic link between BPA and T2DM were also included. Additionally, studies that reported associations between BPA and metabolic factors that may predispose to T2DM, such as changes in insulin sensitivity, insulin resistance, β-cell morphology or β-cell function were also included. Studies investigating critical windows of exposure to BPA were considered valuable as they provide a better understanding of the possible functional consequences of BPA exposure. Furthermore, handsearching and citation review of relevant studies were also conducted to identify studies that were not captured by electronic database search.

### 2.2. Exclusion Criteria

Studies were excluded if they (1) did not directly assess the relationship between BPA exposure and the development of T2DM or a related metabolic parameter—specifically, articles investigating associations with T1DM and/or gestational diabetes as the primary endpoint were not included. This is because GDM typically describes glucose intolerance diagnosed during gestation that resolves after pregnancy. Key articles on GDM as an early life exposure are discussed separately; (2) articles that were not written in English and (3) reviews, editorial letters, case series or case reports, comments.

### 2.3. Data Extraction

Two investigators independently screened article titles and abstracts and performed study selection and data extraction according to the inclusion and exclusion criteria outlined above. Any discrepancies were adjudicated and resolved by a third independent reviewer. For epidemiological studies, the following information was extracted from the selected articles: (1) primary author and year of publication; (2) type of study; (3) study population characteristics, including cohort size, age, ethnicity and gender proportions; (4) samples used to ascertain BPA levels; and (5) main study outcomes. For animal and in vitro functional studies, the main findings of each article were grouped into sections, depending on (1) the organ or tissue being studied; (2) the window of exposure, namely perinatal or adult exposure; and (3) a detailed description of the cellular mechanism implicated.

## 3. Results

### 3.1. Epidemiologic Studies Investigating the Association between BPA Exposure and T2DM

The literature search yielded 27 key epidemiological studies that assessed the relation between BPA exposure and T2DM. These were selected for full-text review. These studies capture a wide range of heterogeneity with regards to epidemiologic design, sample size, ethnicity, and geographical location, as well as the assay methodology used to ascertain BPA levels. Most studies assayed spot urine BPA using liquid chromatography with tandem mass spectrometry (LC–MS/MS) platforms, although some studies have quantified serum BPA with sandwich enzyme-linked immunosorbent assay (ELISA) methods [31,32,33]. Free BPA is generally not detectable in blood samples in the general population as following oral ingestion it is rapidly metabolised and eliminated in urine as the glucuronide conjugate [34,35]. 

#### 3.1.1. Critical Overview of Epidemiologic Studies

A systematic summary of these studies and their salient findings is shown in Table 1**.** Most of the studies selected are characterized by a cross-sectional study design, although case-control cohort and longitudinal studies were also included. Evidence from individual epidemiological studies is generally inconclusive or inconsistent, with conflicting findings being frequently reported. Some studies have reported positive associations for specific subgroups of age or gender [36]. Three meta-analysis have provided pooled estimates of epidemiological data on BPA and T2DM risk [13,37]. These meta-analyses have demonstrated a positive association between increasing BPA levels and insulin resistance or T2DM risk. A summary of the meta-analyses is shown in Table 2.

It is thus unclear whether exposure to BPA is a risk factor for the development of T2DM based on epidemiological examination alone. Several factors might partly account for these conflicting findings. Possible important confounders include the extensive diversity in population ethnicity, lifestyle, dietary intake, use or consumption of BPA-containing products and variation in sample size across investigations. Furthermore, differences in T2DM diagnostic criteria exist between studies. Some studies rely solely on the self-reported diagnosis of diabetes mellitus, while others on an oral glucose tolerance test (OGTT) or HbA1c [38,39].

Additionally, few studies have directly observed a link between BPA and the development of insulin resistance, which is considered a key component in the pathophysiology of T2DM. These factors are exacerbated by inconsistency in study end points and considerable uncertainty with regard to the assessment of BPA exposure. The scarcity of prospective studies also limits the direct evaluation of the cause–effect relationship between specific endocrine disruptors and the incidence of diabetes or related metabolic complications, independent of the traditional risk factors [40]. Cross-sectional and case-control studies are not designed to determine causality between BPA and T2DM, and cross-sectional studies are prone to selection bias which could give rise to a heterogeneity of findings [13,32,41]. 

#### 3.1.2. BPA Dose–Response Relationship

BPA concentrations in human blood (serum and plasma) are in the range of 0.3–4.4 ng/mL (1.3–19.4 nM) in developed countries [42]. The relationship between BPA and metabolic outcomes is complicated by evidence from studies which suggests that it exhibits a non-monotonic dose–response curve. Few substances exhibit such a pattern, and regulatory bodies therefore tend to assume linear relationships when calculating the safe range of doses of a particular substance [43]. Epidemiological studies also assume a linear relationship between BPA exposure and T2DM risk, although this may not reflect the actual dose–response relationship [13]. Most articles discussed in this review report adverse effects at concentrations of BPA that are much lower than the tolerable daily intake proposed by the US Environmental Protection Agency (50 µg/kg–bw/day) and the European Food Safety Authority (4 µg/kg–bw/day). This implies that excessively high doses of BPA may not capture the full extent of its effects on the body. Several studies have described adverse effects of BPA at doses below the calculated safe dose [44,45]. The non-monotonic dose–response of BPA is likely due to its effects on receptor kinetics and specificity [32,46,47,48,49]. Moreover, xenoestrogens such as BPA can act synergistically with endogenous physiological oestrogen, meaning that lower doses of BPA can be sufficient to disrupt certain endocrine features [50]. Given the quantity of evidence that supports a non-monotonic dose–response curve for BPA, further research needs to be performed in order to revise the range of exposure considered to be safe, and to better explain the molecular mechanism underlying such a response.

#### 3.1.3. Assessing BPA Exposure—Challenges and Caveats

Pharmacokinetic studies demonstrate that BPA and the related phthalates are rapidly metabolised, and have half-lives of less than 24 h in the human body [14]. Spot measurements at single time points do not capture chronic exposure to BPA, as evidenced by studies showing a low correlation of serial urinary BPA measurements over 6-month periods [51,52]. Thus, the single measurement of urinary BPA in most studies reflects recent BPA exposure and limits its biological interpretation [53]. Assessing BPA levels in a single spot urine sample at the time of T2DM diagnosis is a poor proxy marker of exposure in the years preceding diagnosis. As a weak endocrine disruptor, BPA has the potential to exert deleterious metabolic effects during specific ‘exposure windows’ in early development, with the consequent onset of metabolic disease in later adulthood. Additionally, urinary BPA levels vary diurnally and can be affected by diuretic therapy [7]. Studies aimed at reducing dietary BPA consumption were not successful at lowering total body BPA levels, suggesting that other sources of exposure exist [11]. The ubiquitous sources of exposure also restrict the ability of investigators to identify study participants that are completely unexposed to environmental BPA. 

The long-term effects of short-lived EDCs should be thus evaluated by the incorporation of both baseline and serial assessments of BPA levels in longitudinal studies in relation to metabolic outcomes. Studies have recently applied a time-averaged statistical approach to derive the cumulative average exposure to BPA [54]. To further assess chronic exposure, the health outcomes in occupationally exposed individuals can be compared to environmentally exposed individuals. Selected occupations (BPA/epoxy resin manufacture and thermal paper contact) result in significantly higher BPA levels, and a multitude of adverse outcomes, including reproductive and endocrine effects, have been described in occupationally exposed individuals [55].

Large biomonitoring studies have also provided evidence that BPA levels do not decline rapidly with fasting time, suggesting that non-dietary exposure or accumulation in body tissues are relevant in BPA pharmacokinetics [56]. BPA is lipophilic, with a fat: blood coefficient of 3.3 [57]. Additionally, the transdermal and sublingual absorption of BPA that bypasses first-pass metabolism have been documented [58,59]. Clearly, reliance on single spot measurements of urinary BPA provides no evidence of bioaccumulation, due to the potential for the sequestering of EDCs in other body compartments.

A large number of human biomonitoring studies using tissue, blood and urine indicate clearly that the general population is exposed to BPA [12]. BPA is routinely detected in blood in the unconjugated form (nanograms per millilitre range) while conjugated BPA is routinely detected in urine. Unconjugated (parent) BPA is the biologically active EDC. BPA conjugates do not bind nuclear oestrogen receptors, although other biological activities have been attributed to modified BPA [60,61]. The measurement of serum/blood BPA has thus attracted considerable interest in the scientific community. Studies assessing BPA in the serum or blood largely use analytical chemistry methods, although ELISA and radioimmunoassay techniques have been published (reviewed by Vandenburg et al. [12]). Although they are typically smaller in magnitude than urinary BPA studies, most serum or blood BPA studies have reported concentrations of unconjugated BPA in the low ng/mL range. Such concentrations are identical to BPA conjugates in urine and indicate internal exposure to biologically active BPA. Importantly, some investigators have failed to detect BPA in blood samples [62,63].

It is essential to emphasise that the measurement of unconjugated BPA in blood has been steeped in controversy. Several issues have been raised that dispute the validity of such measurements, ranging from the contamination of reagents with BPA, the leaching of BPA during sample collection and storage, and the deconjugation of BPA metabolites during sample extraction, thus leading to the erroneous overestimation of free BPA [64,65]. A detailed overview of the controversy concerning unconjugated bioactive BPA in human serum is beyond the scope of this review. However, key industry and FDA-funded studies have rejected biomonitoring data reporting unconjugated BPA in human blood based on post-exposure contamination [66,67]. Conversely, a round-robin study recently showed that the accurate and sensitive quantification of serum unconjugated BPA can be achieved in multiple laboratories, and that BPA contamination is not ubiquitous [68]. Similarly, other authors have disputed universal contamination in serum BPA assays [69].

Notwithstanding this controversy, the direct determination of both total and unconjugated serum BPA can be considered an asset in studies investigating association and causality between BPA and T2DM. More epidemiological studies should endeavour to quantify serum BPA, given its physiological relevance arising from its direct action on nuclear steroid receptors. Urinary BPA levels do not provide meaningful information on internal bioactive concentrations or exposure routes. 

The studies summarised in this review article are characterised by considerable variability in the laboratory methodology used to quantify BPA and its reporting. Of note, only some studies specifically report on the use of a comprehensive quality control system to ensure that samples were not contaminated during handling, storage or analysis [7,40,53,70,71,72,73]. Other studies solely report the limits of detection and within and between day precision metrics [47,74,75]. Studies using ELISA employ the use of kits from different manufacturers and report variation in intra and inter-assay precision [9,32,33]. It is likely that variability in these analytic parameters limits the direct comparison of epidemiological studies. 

#### 3.1.4. BPA Analogues and Co-Exposure

In addition to the challenges presented by the short biological half-life of BPA, its rapid first-pass metabolism and possible contamination during sampling or analysis, BPA substitutes pose further difficulty. BPA substitutes include structural analogues bisphenol S (BPS) and bisphenol F (BPF) that are increasingly used by manufacturers in BPA-free products. Several key differences between BPA and its substitutes exist. BPS is more stable, less lipophilic and less cytotoxic than BPA [76,77]. Although they have been touted as potentially safer alternatives to BPA, current studies suggest they exert similar endocrine disrupting effects. BPA substitutes possess hormone activity and are increasingly detected in human urine [78,79,80]. Studies have reported adverse health effects linked to BPA alternatives, and have shown that BPS and BPF have similar potency and action as BPA [81,82]. Humans are concurrently exposed to multiple xenobiotics through dietary and environmental sources, and exposure to BPA is thus a likely an indicator of exposure to multiple EDCs. At the population health level, the adverse effects of mixtures of EDCs are more relevant than those of single substances. Recently, Park et al. showed that mixtures of BPA, BPS and BPF exert oestrogen agonist and anti-androgen activities at lower concentrations than single bisphenols [83]. Hence, the mixture effect is a likely confounder variable in epidemiological studies exploring the BPA–T2DM link. 

### 3.2. BPA Exposure and Early Life Factors 

Major changes in glucose regulation accompany pregnancy, characterised by a degree of insulin resistance, and increased maternal glucose secretion. Commonly, gestational diabetes develops if insulin secretion does not increase sufficiently to counteract the insulin resistance. This can lead to further maternal and foetal metabolic complications, including an increased risk of T2DM in later life [90,91]. The effects of EDCs in pregnancy have been widely studied. Studies have shown that BPA can be passed on from mother to offspring via the placenta and breast milk [12]. Since oestrogen plays an essential role in regulating maternal adaptations to pregnancy, xenoestrogens, such as BPA, are expected to interfere with these processes. In this context, pregnancy and the post-partum state can be considered a critical window of exposure where EDCs can cause significant harmful effects on both mother and foetus.

Pregnant mice exposed to 10 µg/kg/day of BPA throughout gestational days (GD) 9–16 demonstrated glucose intolerance. A clear dose–response relationship was not described, as the administration of higher quantities of BPA (100 µg/kg/day) did not lead to a greater degree of glucose intolerance, further hinting at a non-monotonic response. Both BPA doses caused hyperinsulinaemia, whilst 10 µg/kg/day BPA disrupted hepatic insulin signalling and enhanced insulin resistance. While the lower BPA dose resulted in detrimental effects during pregnancy, the higher dosage elicited more deleterious effects post-pregnancy. These include increased post-partum weight gain, hyperinsulinaemia, and elevated levels of plasma leptin, triglycerides and cholesterol [90]. Furthermore, mice developed impaired glucose tolerance and decreased insulin sensitivity at four months postpartum. In contrast, nonpregnant female mice treated with the same doses of BPA did not show any significant changes in glucose tolerance and insulin sensitivity [91]. Importantly, the metabolic changes triggered by BPA in pregnant mice seemed to resolve after delivery but reappeared later in life. This implies that foetal BPA exposure during gestation has long term irreversible effects on the risk of metabolic disorders in later adulthood. Exposure to BPA during gestation also resulted in a significant reduction in maternal glucose-stimulated insulin secretion (GSIS) several months after delivery. BPA exposure triggers changes in the β-cell life cycle, with increased apoptosis and decreased proliferation leading to a reduced β-cell mass. These effects are at least partly due to the decreased expression of certain cell cycle activators, such as cyclin D2 (*CCND2*), and the increased expression of some cell cycle inhibitors, such as cyclin dependent kinase inhibitor 2A (*CDKN2A*). Such alterations in the β-cell mass are thought to arise due to the oestrogen-mimetic effects of BPA [91]. 

The offspring of mice exposed to BPA during gestation similarly exhibited disordered metabolism. Decreased insulin sensitivity and increased GSIS were detected within 6 months of birth. Of note, males were more adversely affected than females, and lower doses of BPA had more unfavourable effects than the higher dose. It is likely that female offspring are less affected than males as oestrogens within the physiological range may protect against diabetes. These metabolic alterations in the offspring of mice exposed to BPA during gestation result in a higher risk of developing T2DM and associated disorders later in life [90]. Similarly, Manukyan et al. showed that gestational exposure to very low doses of BPA via the oral route leads to insulin hypersecretion in rat offspring up to one year after exposure. They demonstrated that at lower doses of BPA, GSIS was enhanced in pancreas from both 5- and 52-week-old offspring. Contrastingly, at higher doses of BPA a reduced GSIS was observed. These findings further emphasise the long-lasting effects of BPA, even after exposure is terminated. The pattern of insulin secretion induced by higher BPA doses is quite similar to that observed in diabetic patients, where first-phase insulin secretion was reduced. Additionally, the lower dose of BPA could also be detrimental as the prolonged hypersecretion of insulin may cause added stress to the β-cell [92]. 

Since obesity is a major risk factor for T2DM, measurements of weight gain and fat distribution in early life can be predictive of possible metabolic complications in later life. Wei et al. observed the aftermath of perinatal BPA exposure on offspring fed either a standard diet or an high-fat diet (HFD). The offspring of dams exposed to BPA exhibited higher weight gain and larger adipocytes at 19 weeks of age compared to controls, an effect that was exacerbated by HFD. Male mice demonstrated greater disruptions in serum lipid levels than females, primarily featuring higher triglyceride and lower HDL cholesterol levels. Moreover, fasting blood glucose and insulin levels were increased by week nine, and this was accompanied by islet changes—including enlarged and scattered β-cells in the HFD group. In addition, several transcription factors were downregulated. Of note, the standard diet group and HFD females showed higher GSIS, while HFD males showed lower GSIS. These findings indicate the faulty development of β-cells early on in life, which is aggravated by an HFD. Surprisingly, very high doses of BPA (250 and 1250 µg/kg–bw/day) did not induce any significant changes [93]. 

Evidence from epidemiological studies suggests that low birth weight leads to rapid post-weaning weight gain, a phenomenon termed “centile crossing”. This is a possible risk factor for the subsequent development of obesity and T2DM. A study by Taylor et al. demonstrated this effect in newborn male mice. The authors showed that mice with the highest post-weaning weight gain exhibited the greatest elevation of blood glucose after a low-dose glucose tolerance test. Furthermore, perinatal BPA exposure impacts on body weight at weaning as well as the rate of post-weaning growth. The findings from this study suggest that foetal exposure to low-dose BPA is more likely to affect a sensitive subpopulation characterised by early life growth restriction and rapid catchup growth post weaning, which could be at higher risk for the development of T2DM [94]. 

Other studies have shown that the metabolic outcomes of BPA differ according to the timing of the exposure window, as well as dosage and gender. Liu et al. show that mice exposed to BPA during the postnatal period showed the highest weight gain, while those exposed in early pregnancy weighed less than controls [95]. β-cells are susceptible to the effects of BPA in both the foetal and neonatal periods, resulting in increased β-cell mass with reduced or invariable insulin secretion. Similar detrimental effects of BPA exposure were observed in larger mammals such as sheep. The offspring of sheep exposed to BPA during gestation demonstrate pre-pubertal hyperglycaemia and insulin resistance, an identical phenotype to T2DM. This prepubertal insulin resistance sets the stage for further disruption in glucose and insulin regulation after puberty [96], as was observed in mice from studies discussed earlier [90,91]. Furthermore, sheep exposed to BPA demonstrate a higher visceral to subcutaneous adipose tissue ratio and increased CD68 expression, a marker of inflammation indicative of macrophage infiltration in adipose tissue [96].

Several investigators have investigated birth outcomes following BPA exposure. The Korean Mothers and Children’s Environmental Health (MOCEH) study revealed significant positive associations between maternal BPA exposure and higher birth weight, which were stronger in male neonates [97]. A recent meta-analysis of seven independent studies showed a similar direction of association between BPA and higher birth weight [98]. Prenatal BPA exposure has also been linked with sex-specific changes in hypothalamic–pituitary–adrenal axis function, and the epigenome-wide methylation analysis of cord blood DNA similarly identified sex-specific effects [99,100]. Importantly, a prospective study showed that prenatal BPA exposure has residual sex-specific effects on glucose and lipid metabolism that persist into childhood, although maternal bisphenol exposure was not associated with childhood adiposity [101,102]. Bansal et al. also showed that maternal BPA exposure exerts gender-specific multigenerational effects, impairing insulin secretion in male but not female mice through epigenetic changes involving insulin-like growth factor 2 (*IGF2*)—a key β-cell gene [103]. Sexual dimorphic effects of prenatal BPA exposure on hepatic gene expression, body composition and glucose homeostasis in murine and rat models have been reported [104,105]. These findings further support the role of ER-dependent mechanisms in mediating gender-specific metabolic effects.

In summary, the foetal and neonatal periods constitute key critical intervals for BPA exposure. An overview of the adverse metabolic effects of BPA at these life stages is shown in Figure 1. The ongoing pancreatic development can result in the programming of metabolic outcomes that lead to insulin resistance and the development of glucose intolerance in adult life. Different window of BPA exposure can thus have varying effects that depend, in part, on the offspring’s developmental stage at exposure. Further research is required to uncover the precise mechanism by which BPA disrupts the delicate programming of this part of the endocrine system [95]. 

### 3.3. BPA and T2DM—From Epidemiological Associations to Mechanistic Evidence

Several in vitro and in vivo animal studies have attempted to unravel the mechanistic link between BPA and T2DM. The findings from key functional studies are outlined next.

#### 3.3.1. Effect of BPA on Glycaemia, Insulin Resistance and Lipids

As BPA is a phenolic xenoestrogen pollutant, it imitates the effects of endogenous oestrogen (E_2_). Multiple studies have shown that both BPA and E_2_ induce a rapid dose-dependent change in glycaemic response and insulinaemia in adult mice [16,106,107]. Alonso-Magdalena et al. demonstrated that acute exposure to a single low dose of either BPA or E_2_ produces a rapid decrease in the rise of glycaemia within 30 min of the first injection. This is the result of an increase in plasma insulin. Furthermore, sustained exposure to either BPA or E_2_ at 10 and 100 μg/kg/day resulted in higher β-cell insulin levels, an effect mediated through the oestrogen receptor. Exposure to the higher BPA dose resulted in hyperinsulinemia, with plasma insulin levels comparable to those in late pregnancy. Both the E_2_ and the BPA-treated mice exhibited 1.7- and 1.53-fold higher circulating insulin levels, respectively, with unvarying blood glucose levels. This is characteristic of insulin resistance [107]. The hyperinsulinaemic effect of BPA was also demonstrated by murine studies showing that eight days of BPA exposure resulted in an increase in GSIS and inhibition of basal insulin secretion. The effects of BPA on glycaemia and GSIS were completely abolished in islets from ERα knockout mice but not in ERβ knockout mice, suggesting that BPA upregulates pancreatic insulin content through a mechanism involving the activation of ERα [15]. Taken together, these findings provide strong support for a direct link between BPA, β-cell function and insulin resistance [15,108,109].

Ding et al. investigated the effects of long-term exposure to a dose (50 µg/kg/day) of BPA and its interaction with high-fat diet. They reported that serum glucose levels were significantly higher in BPA-treated male rats than the untreated controls following 35 weeks of BPA exposure. BPA treatment in combination with a high-fat diet (HFD) resulted in a higher elevation of serum glucose levels than a HFD alone. The study also showed that BPA-treated rats on a standard diet had an increase in serum insulin levels, an elevated homeostatic model assessment of insulin resistance (HOMA-IR) and a reduced insulin sensitivity index (ISI) compared with the control group. The persistent hyperglycaemia, despite higher circulating insulin levels elicited by BPA, is an indicator of insulin resistance and disrupted glucose homeostasis [110]. 

In a similar study, Marmugi et al. administered increasing doses of BPA to six-week-old male mice for 32 weeks via drinking water. Following long-term exposure, a dose-dependent elevation of plasma glucose levels was observed. Moreover, the mice exposed to the highest dose showed significantly impaired glucose tolerance when compared to the control group. In contrast with the previously mentioned studies, this study did not observe a significant difference between the plasma insulin levels of BPA-treated mice and the control group [111]. Similarly, Moon et al. reported that the 12-week exposure to 50 µg/kg/day of BPA via the oral route to mice on HFD resulted in glucose intolerance and insulin resistance. In keeping with Marmugi et al., no statistically significant increase in fasting serum insulin levels between the BPA-treated group and the control group was detected [112]. 

An HFD increases serum triglycerides and total cholesterol levels when compared with the standard diet; however, no significant difference in lipid profiles was observed between the BPA-treated groups and the untreated groups [110]. Conversely, Marmugi et al. demonstrated an increase in total cholesterol levels in mice exposed to BPA. This effect is mediated by the upregulation of hepatic genes involved in de novo cholesterol biosynthesis [111]. 

#### 3.3.2. Effect of BPA on β-Cell Mass, Morphology and Function

When compared to the other phenolic oestrogen pollutants such as diethylstilbestrol, octylphenol and nonylphenol, BPA has a lower oestrogenic activity [113]. In rat insulinoma cell lines, 48 h exposure to BPA decreases cell viability, disrupts GSIS and triggers apoptosis in a dose-dependent manner [114]. BPA activates β-cell apoptotic signalling via the increased expression of pro-apoptotic Bax protein and the reduced expression of anti-apoptotic Bcl-2. Structural defects in rat β-cell mitochondria that precede changes in glucose homeostasis have been documented following the administration of 50 μg/kg/day of BPA [93]. Using rat islet cells, Song et al. showed that at a dose of 2.5 µg/L, BPA reduces β-cell viability, whereas an identical effect was observed with much higher doses of the other oestrogenic pollutants. This is of interest as 2.5 µg/L is within the range of human exposure to BPA. BPA affects the insulin-secreting ability and mass of isolated islets. β-cell diameter increases at BPA doses between 2.5 and 25 µg/L but decreases at a dose of 250 µg/L, possibly due to cytotoxic effects on the β-cells at a high dose. Transmission electron microscopy (TEM) studies revealed that 25 µg/L of BPA results in a reduced amount of filled insulin vesicles and an increase in empty vesicles [115]. Similarly, prolonged BPA exposure causes an increase in β-cell mass due to islet expansion, an effect that is aggravated by HFD [110]. Conflicting findings have also been described. Moon et al. reported no change in the islet morphology in mice exposed to both BPA and HFD when compared to mice given a HFD only. Electron microscopy showed no difference in the number and shape of mitochondria, as well as the insulin content of the cells [112]. The heterogeneity of findings can be partly accounted for by differences in rodent species, their age, duration, route, and level of exposure.

As many EDCs follow a non-monotonic dose–response curve, the most effective dose of EDC may not necessarily be the highest one. Instead, the focus is being directed towards the dose that has maximal adverse effects on a specific metabolic pathway [111]. An inverted U-shaped relationship exists between incremental doses of BPA and GSIS. A dose of 0.1 µg/L of BPA was enough to cause a significant increase in GSIS, whilst doses of 25 and 250 µg/L caused a significant decrease [115]. Lower doses of BPA, mainly 0.1 µg/L and 1 µg/L, have also been shown to increase both basal and GSIS [116]. As GSIS depends on signals generated by β-cell mitochondria, any mitochondrial abnormality could be a potential contributor to metabolic disorders such as T2DM [117]. Song et al. showed that BPA and other phenolic oestrogens induce ultrastructural changes in β-cells. Specifically, β-cell mitochondrial swelling with a loss of structural integrity, impaired mitochondrial cytochrome c oxidase function and reduced cytosolic ATP levels were observed in BPA-treated islets [115].

Alternative mechanisms linking BPA to β-cell apoptosis and a decrease in β-cell mass have been described. BPA is implicated in β-cell damage through its interaction with human islet amyloid polypeptide (hIAPP). hIAPP is a 37-residue soluble polypeptide that is produced by the β-cell and is co-secreted with insulin. The function of hIAPP in the β-cell is not fully understood, and distinguishing physiological from pathological effects is a considerable challenge [118]. Physiologically, it contributes to glycaemic regulation by inhibiting insulin and glucagon secretion, inhibiting gastric emptying, as well as acting centrally to induce satiety [118,119,120]. hIAPP monomers also possess intrinsic propensity to misfold, forming β-sheet oligomers that assemble into linear fibrils. The oligomers and fibrils exert cytotoxic effects on pancreatic β-cells by inducing membrane permeabilisation and disruption [121,122]. The link between hIAPP and T2DM has been well established by studies demonstrating that IAPP aggregates are detectable in most diabetics and a spatial correlation exists between IAPP deposits and loss of β-cell mass [118,121,123]. Aggregates of IAPP insert into the β-cell membrane, causing the leakage of cellular contents and eventually apoptosis. Critically, in vitro studies using a rat insulinoma cell line demonstrated that BPA acts in a dose-dependent manner to promote hIAPP aggregation and membrane disruption [124]. As the cell membrane becomes more permeable, Ca^2+^ ions enter the cell and trigger the generation of harmful reactive oxygen species (ROS). This process was observed in the presence of BPA and hIAPP, but not with BPA alone, hence suggesting the possible cooperative action of the two molecules [124]. This study by Gong et al. thus showed that BPA can contribute to T2DM development through the modulation of β-cell survival. The highly fibrillogenic hIAPP is normally protected from aggregation by its interaction with inhibitors of fibril formation, such as insulin and zinc ions [125]. Some authors have hypothesised that the formation of IAPP aggregates plays a critical role in the transition from early-stage insulin resistance to overt T2DM. The compensatory hyperinsulinemia arising due to insulin resistance leads to a higher production of hIAPP and increases endoplasmic reticulum stress, which exacerbates β-cell demise, in turn leading to greater insulin/IAPP production in the surviving β-cells [126,127]. The mechanisms inducing the formation of cytotoxic hIAPP aggregates in pancreatic β-cells, their role in the development and progression of T2DM, and the significance of BPA in this process is, however, not fully resolved.

#### 3.3.3. Effect of BPA on β-Cell Gene Expression

The effect of BPA on β-cell function and impaired insulin secretion is a consequence of dysregulated β-cell gene expression. Twenty-four-hour exposure to 25 µg/L BPA results in a downregulation of the pancreatic glucose transporter (*SLC2A2*) and glucokinase (*GCK*), and consequently reduced insulin secretion. Glucokinase is an enzyme involved in the first step of glycolysis, as it catalyses the phosphorylation of glucose to glucose-6-phosphate. The decreased expression of *GCK* and *SLC2A2* is the result of downregulation of key β-cell genes, including insulin promoter factor 1 (*PDX1*), and hepatocyte nuclear factor 1A (*HNF1A*). The synaptosome-associated protein of 25 kDa (*SNAP25*) expression is also decreased in response to BPA. This gene functions in vesicle fusion to plasma cell membranes during exocytosis [115]. 

Chronic BPA exposure, in isolation or in combination with HFD, elicits cellular stress responses. The upregulation of genes in cellular autophagy pathways, such as microtubule associated protein 1 light chain 3 alpha (*MAP1LC3A*) and beclin 1 (*BECN1*) has been documented in response to BPA [110]. Similarly, BPA exposure triggers the altered expression of genes encoding molecular chaperone proteins, such as the heat shock protein 90 beta family member 1 (*HSP90B1*) and heat shock protein family A (Hsp70) member 5 (*HSPA5*). The observed transcriptomic changes are dependent on the prevailing glucose concentration, the duration and concentration of BPA exposure and thus requires further evaluation [116]. 

#### 3.3.4. Cellular Mechanisms of BPA Action

Recent studies have highlighted the role of BPA in modulating the expression and function of β-cell ion channels that regulate GSIS [128]. In the fasting state, the consumption of ATP by cellular metabolic pathways results in low ATP:ADP ratio, an increased activity of ATP-sensitive K^+^ channels and hyperpolarised β-cell membrane potentials. Following glucose uptake by β-cells in the postprandial state, the rise in ATP:ADP ratio results in the generation of an oscillatory current that alternates between hyperpolarised and depolarised phases. The depolarisation phase is mediated by the activation of voltage-activated Ca^2+^ and Na^+^ channels and leads to bursts of action potentials, while the repolarisation phase is mediated by the activation of small-conductance Ca^2+^-activated K^+^ channels (K_Ca_2.3 channels). The alternating depolarisation and hyperpolarisation phases cause an oscillatory release of Ca^2+^, which in turn, stimulates the exocytosis of insulin-containing vesicles from the β-cell [16,128].

Martinez-Pinna et al. showed that the exposure of murine β -cells to BPA results an imbalance in their electrical activity. Electrophysiological studies on islets from mice exposed to BPA demonstrate a decrease in Na^+^ and K^+^ currents. Microarray gene expression analysis identified the dysregulation of several genes encoding components of Na^+^ and K^+^ channels, including sodium voltage-gated channel alpha subunit 9 (*SCN9A*), potassium voltage-gated channel subfamily B member 2 (*KCNB2*), potassium calcium-activated channel subfamily M alpha 1 (*KCNMA1*) and potassium voltage-gated channel interacting protein 1 (*KCNIP1*). These effects were not observed in β-cells from oestrogen receptor β knockout mice (ERβ^−/−^), suggesting that the modulation of Na^+^ and K^+^ currents by BPA are mediated by ERβ [128]. Importantly, the concentration of BPA used in this experiment is identical to that in human serum, and changes in the expression or activity of beta cell ion channels are implicated in T2DM [129]. Such changes in β-cell electrical activity may contribute to impaired insulin secretion and the diabetogenic effects of BPA [128]. Conversely, in human studies, oral BPA supresses second-phase GSIS in obese individuals [49]. Possibly, BPA has differing effects on the mechanisms of the early response (release of stored insulin) vs. later insulin response (de novo insulin synthesis). Other murine studies have supported this concept, showing that the effects of BPA on insulin exocytosis and secretion vary according to the absence or presence of glucose and its concentration, and are mediated by both ERβ and ERα [130]. 

Soriano et al. further demonstrated the role of ERβ in the regulation of the K_ATP_ channel activity, Ca^2+^ flow and insulin release elicited by environmentally relevant doses of BPA. BPA reduced K_ATP_ channel activity by almost half in islets from both mice and humans and increased the frequency of pulsatile Ca^2+^ release and insulin secretion. These effects were absent in ERβ^−/−^ mice, confirming that BPA acts through ERβ to induce changes related to insulin secretion. It is possible that BPA can act in this manner in humans as the concentration of BPA in serum is identical to the dose used in this study. Further research is required to fully understand the involvement of other oestrogen receptors in β-cells, as these may act in a non-classical manner when bound to BPA. In addition to animal models, more human β-cell studies are needed to account for any differences between rodent and human physiology [16]. 

#### 3.3.5. Effect of BPA on Other Tissues

Exposure to BPA in animal studies has been shown to induce significant alterations in whole body metabolism and feeding patterns. Animal studies have described sex-specific alterations in hypothalamic pathways controlling the energy intake and expenditure induced by BPA [131]. Reduced proopiomelanocortin expression, increased leptin and neuropeptide Y levels in the arcuate nucleus have been reported, suggesting that BPA induces the dysregulation of the melanocortin system that regulates energy balance [132]. Furthermore, when compared to controls, BPA-treated mice exhibited unchanged diurnal but reduced nocturnal food intake, a decrease in spontaneous locomotor activity and a decrease in overall energy expenditure. Potentially, these effects are accounted for by the oestrogenic properties of BPA that allow it to act on the central nervous system to induce changes in leptin and insulin levels [108]. 

Skeletal muscle carries a high metabolic demand and is a major insulin-sensitive tissue. It therefore plays a critical role in blood glucose homeostasis. Insulin-signalling pathways in muscle can also be altered by exposure to environmentally-relevant doses of BPA [108]. Insulin receptor substrate 1 (IRS1) is essential for various processes, including insulin receptor signalling, the growth of myofibers, insulin-dependent glucose uptake and glycogen synthesis [133]. Exposure to BPA results in the upregulation of IRS1 in skeletal muscle. In BPA-treated mice, the insulin-stimulated phosphorylation of the insulin receptor β subunit is impaired, resulting in a reduction in insulin-induced Akt phosphorylation. These alterations in insulin receptor signalling pathways in response to short-term exposure to low doses of BPA provide insight into the development of peripheral insulin resistance, a key hallmark of T2DM [108]. Of note, similar observations in Akt phosphorylation were recorded in studies using muscle cells from T2DM patients [134]. 

In hepatocytes, BPA elicits identical perturbations in insulin signalling pathways to those observed in skeletal muscle [108]. In the liver, IRS1 participates in the regulation of lipid and glucose metabolism, and this protein is upregulated by exposure to BPA under basal conditions [133]. Evidence also points to the disruption of hepatic glucose regulation by BPA. Acute (single dose) and chronic (over 2 weeks) BPA exposure results in a significant reduction in glucokinase activity under several physiological glucose concentrations. While oestrogen stimulates glucokinase activity, BPA treatment reduces it under these conditions [135]. Conversely, 500 µg/kg/day BPA exposure over eight months lead to an increase in glucokinase and pyruvate kinase enzymes. BPA also induces oxidative stress in hepatocytes by reducing the activities of antioxidant enzyme superoxide dismutase, glutathione peroxidase and catalase [136]. Furthermore, animal studies have shown that BPA exposure is linked to changes in liver activity, hepatocyte DNA damage and the induction of hepatocyte apoptosis [137,138,139]. In the liver, BPA stimulates lipid accumulation through the upregulation of lipogenic genes, such as sterol regulatory element binding protein 1 (*SREBP1*) [140,141]. This evidence suggests that BPA contributes to the development of non-alcoholic fatty liver disease (NAFLD), a frequent metabolic disturbance in T2DM [142,143]. 

In addition, the sustained exposure of adult mice to BPA over a period of eight months results in the significant upregulation of genes involved in de novo lipogenesis. These include *FASN*—the gene encoding fatty acid synthase, a principal enzyme functioning in the de novo synthesis of long-chain fatty acids, as well as thyroid hormone responsive protein (*THRSP*), syndecan 1 (*SDC1*), patatin like phospholipase domain containing 3 (*PNPLA3*) and sterol regulatory element binding transcription factor 1 (*SREBF1*). Moreover, such long-term BPA exposure leads to an increase in the key enzymes of de novo cholesterol biosynthesis, including 3-hydroxy-3-methyl-glutaryl-coenzyme A reductase (HMG-CoA reductase). Increased hepatic cholesterol levels have been reported in mice exposed to BPA, with no significant change in the levels of hepatic triglycerides and cholesteryl esters [111]. The expression of these genes is also increased by hyperinsulinemia, and the observed metabolic deregulation is also consistent with evidence from early-life exposure animal studies. Conversely, it contrasts with the proposed reduction in insulin secretion suggested by some studies, although this is explained by the differing effects of BPA on early and late phases of insulin secretion. A summary of the general metabolic effects of BPA is illustrated by Figure 2.

#### 3.3.6. Additional Mechanisms of BPA Action—Beyond Oestrogen Receptors

Most studies have focused on the physiological responses induced by BPA through its binding to nuclear ERα/ERβ subtypes. However, BPA exerts endocrine-disrupting effects through several cellular pathways. BPA binds oestrogen-related receptors (ERRs)—orphan nuclear receptors which are closely related to ERα/ERβ and that possess constitutive transcriptional activity, potentially impacting on the regulation of common target genes [144]. BPA has also been shown to exert both agonist and antagonist effects on thyroid hormone receptor (TR), and it supresses TR-mediated transcription in a dose-dependent manner. [145,146]. BPA competes with 5α-dihydrotestosterone for binding to androgen receptor (AR) to exert anti-androgenic effects [147]. BPA modulates the activity of several transcription factors that regulate energy homeostasis and adipogenesis. Primarily, peroxisome proliferator-activated receptors (PPARs) are nuclear receptors with pleiotropic metabolic effects on skeletal muscle, liver, adipocytes, and the gut. Animal and human studies have shown that BPA induces adipocyte and hepatocyte PPARγ expression, leading to fasting hyperglycaemia and glucose intolerance [90,148]. Other studies have implicated the CCAAT/enhancer-binding proteins (C/EBPs) family of transcription factors in BPA-induced triglyceride accumulation in human adipose-derived stem cells [149]. BPA has been shown to modulate the synthesis of steroid hormones—including testosterone, androstenedione and oestradiol by reducing the expression of rate-limiting proteins in steroidogenic pathways [150]. Recently, the immunomodulatory effects of BPA were highlighted by studies showing that it diminishes leukocyte telomerase activity and accelerates telomere shortening in CD8+ T cells [151].

An additional mechanism of BPA action involves the modulation of epigenetic mechanisms through changes in DNA methylation, histone modification and changes in microRNA expression. Reviews of BPA-induced epigenetic changes have been provided by several authors [152,153]. The transgenerational epigenetic inheritance of changes in glucose homeostasis in animal studies induced by BPA through histone modifications affecting *PDX1* and insulin-like growth factor 2 (*IGF2*) expression have been documented [154,155]. BPA alters hepatic glucokinase promoter methylation, further supporting its role in foetal reprogramming and the subsequent development of metabolic disorders in adulthood [156,157]. 

Clearly, multiple complex cellular pathways are implicated in the modulation of glycaemic responses by BPA, some of which can result in stable heritable epigenetic changes. The integration of these pathways is challenging, as the resulting physiological effects are dose and tissue dependent. Whether the endocrine disrupting effects of different mechanisms of BPA action concur is also debatable. 

### 3.4. Limitations of Animal Studies

Several factors need to be taken into consideration when extrapolating animal studies to humans. Primarily, different rodents can exhibit varying tolerances to the effects of EDCs due to underlying variation in genetic factors. For instance, the F344 rat strain is considered to be more sensitive to oestrogen, while Sprague-Dawley and Wistar rats have much lower sensitivity to oestrogen [92]. Secondly, long-term BPA exposure can produce different effects from short-term exposure, possibly due to changes in metabolism and the age of the animal. Furthermore, a specific dose of BPA can elicit distinct effects in different tissues and even between different metabolic pathways in the same organ [92]. The oestrogen receptor-independent effects of BPA outlined above require further assessment in research models. This necessitates the implementation of specific experimental endpoints to appraise their contribution to BPA-induced changes in glucose homeostasis in both animal models and man. 

While the route of administration of a particular dose of BPA is typically highly controlled in laboratory experiments, this is not the case in humans. Humans are constantly exposed to small quantities of BPA via many routes, rather than a single dose of BPA via a specific route. Since the largest contribution of BPA in human serum is by oral consumption, this is the preferred route of BPA administration in most animal studies. However, recent works have suggested that subcutaneous BPA administration may produce a ratio of conjugated to unconjugated BPA that closely matches that found in human serum [91]. Additionally, variation in pharmacokinetics between rodents and humans is likely to impact on tissue bioavailability of BPA. 

## 4. Conclusions

The extensive body of evidence outlined above provides insight into the multiple mechanisms through which BPA, as a xenoestrogen, modulates physiological pathways linked to the development of T2DM. BPA acts on multiple tissues involved in the regulation of glucose homeostasis. It can positively or negatively modulate pancreatic insulin release and secretion, and alters β-cell gene expression, electrical activity, and β-cell survival. This affects adipocytokine function, regulates hepatic and muscle insulin sensitivity, stimulates de novo lipogenesis, and acts on central nervous system pathways regulating feeding and whole-body metabolism. Nevertheless, it must be emphasised that several caveats exist—primarily the conflicting findings from population-based research, the comparative lack of prospective follow-up studies and the differences between rodent and human physiology that limit the generalisability of findings. In addition, other residual biases—such as variability in BPA exposure and metabolism between individuals and populations—are difficult to account for in large scale epidemiological investigations. The broad pleiotropic effects of BPA coupled with incongruent outcomes in various studies make a comprehensive description of all these challenging effects. Notwithstanding these factors, important lessons can be derived from existing research that can be used to prioritise EDCs on the global public health agenda. It is equally important for clinicians to be aware of the indirect effects of BPA on human development and chronic disease risk. Common disorders such as T2DM have a strong multifactorial aetiology, and this can diminish the effect size of specific variables such as BPA that drive disease development or progression.

## Figures and Tables

**Figure 1 ijerph-18-00716-f001:**
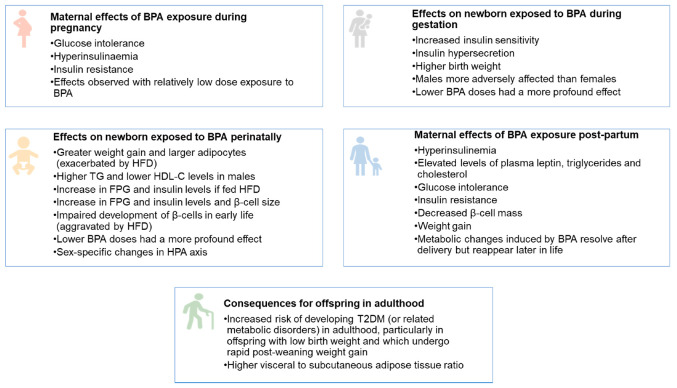
Overview of the adverse metabolic effects of BPA at the foetal, neonatal, and adult stages. HFD—high fat diet. TG—triglycerides. HDL-C—high density lipoprotein cholesterol.

**Figure 2 ijerph-18-00716-f002:**
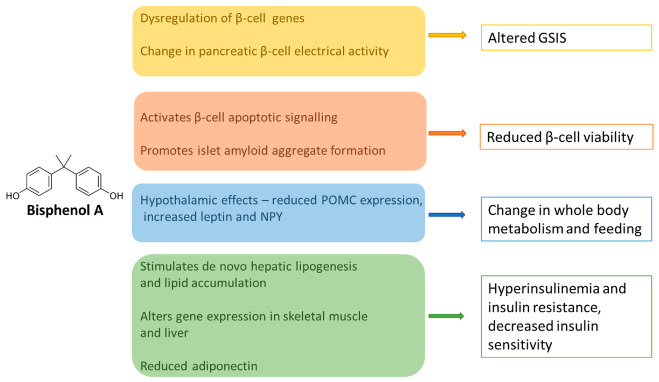
Diagram summarizing the key pathophysiological mechanisms linking BPA to T2DM. BPA acts in pleiotropic ways to impact on the main tissues that regulate glucose homeostasis. In the pancreas, it can modulate insulin production positively or negatively to differentially alter the early and late phases of GSIS. BPA affects insulin sensitivity and insulin receptor signalling in the muscle and liver, increases de novo lipogenesis, reduces the release of adiponectin, and alters central feeding behaviour through multiple pathways. GSIS—glucose-stimulated insulin secretion. NPY—neuropeptide Y. POMC—proopiomelanocortin.

**Table 1 ijerph-18-00716-t001:** Summary of the main findings of the epidemiological studies obtained through the literature search.

Reference	Author, Year	Ethnicity	Main Findings
**Cross-Sectional Studies**			
[53]	Lang et al., 2008	USA	Higher bisphenol A (BPA) concentrations were positively correlated with an increased risk of type 2 diabetes mellitus (T2DM) and obesity. Participants in the highest quartile of BPA level had a greater incidence of T2DM when compared to the subjects in the lower quartiles.
[38]	Ning et al., 2011	China	Median urinary BPA levels did not differ between subjects with normal glucose regulation, impaired glucose regulation, and T2DM. There were more younger participants and males in the highest BPA quartile. Multivariable analysis did not reveal a clear association between BPA levels and T2DM.
[7]	Silver, et al., 2011	USA	This study was carried out in three National Health and Nutrition Examination Survey (NHANES) cycles in consecutive years. The analysis of the total sample showed that a 2-fold increase in urinary BPA was associated with T2DM (Odds ratio (OR) 1.08, 95% C.I.: 1.02–1.16), however this was driven by data from one NHANES cycle.
[41]	Kim and Park, 2013	Korea	Urinary BPA levels were higher in females. No significant difference in mean urinary BPA concentrations between T2DM and normoglycemic individuals was detected.
[11]	Andra et al., 2015	USA	The urinary monochlorinated BPA derivative was significantly associated with T2DM, whereas the parent compound (total BPA) was not.
[9]	Chailurkit et al., 2017	Thailand	A significantly higher level of BPA in impaired glucose tolerance (IGT) individuals compared to normal glucose tolerance (NGT) participants was reported. BPA was associated with IGT but not with T2DM.
[32]	Soundararajan et al., 2019	Asian Indians	Patients with T2DM showed an increased serum BPA level compared to NGT participants. Serum BPA levels in female T2DM patients were significantly higher compared to their respective control individuals. This trend was less significant in males when comparing individuals with T2DM to NGT participants. There was a positive correlation between BPA, poor glycaemic control and insulin resistance.
[70]	Melzer et al., 2010	USA	Elevated BPA concentrations in urine was associated with coronary heart disease and T2DM in pooled data estimates.
[71]	Shankar and Teppala, 2011	USA	Urinary BPA levels were found to be associated with T2DM independently of traditional diabetes risk factors.
[84]	Wang et al., 2012	China	An elevated concentration of BPA was positively correlated to an increased incidence of insulin resistance in adults over the age of 40. This was also accompanied by an increased tendency of generalised and abdominal obesity.
[75]	LaKind et al., 2012	USA	No significant correlation was found between elevated urinary BPA levels and adverse health outcomes. The authors suggest that NHANES datasets are not suitable to draw conclusions between exposure to chemicals with a short half-life, such as BPA, and chronic health conditions.
[73]	Sabanayagam et al., 2013	USA	Elevated urinary BPA levels were linked with prediabetes, after accounting for confounding variables. Subgroup analysis revealed that the association was stronger in females and obese subjects.
[85]	Casey and Neidell, 2013	USA	The study could not draw a definitive conclusion on the association between BPA exposure and chronic disease outcomes from NHANES data as results were sensitive to inclusion/exclusion criteria and the statistical model employed.
[33]	Aekplakorn et al., 2015	Thailand	Serum BPA was independently associated with hypertension in females.
[74]	Beydoun et al., 2014	USA	BPA was associated with an increased β-cell function as well as insulin resistance. The relationship was observed to be stronger in males.
**Case-Control Studies**			
[86]	Piecha et al., 2016	Czech Republic	No association between BPA and T2DM, hypertension, dyslipidaemia, age and body mass index (BMI) was reported.
[46]	Duan et al., 2018	China	Elevated urinary BPA concentrations were associated with an increased risk of T2DM. The relationship was non-linear as the association was significant only in the second and third BPA quartiles but not in the first (lowest) and fourth (highest) quartiles.
[72]	Murphy et al. 2019	Mexico	Higher free urinary BPA in females with self-reported diabetes.
[87]	Li et al., 2018	Saudi Arabia	Mean BPA concentration in diabetic individuals was 3.0-fold higher than in control participants. The association with T2DM was strongest in the third BPA quartile.
[40]	Ahmadkhaniha et al., 2014	Iran	Positive associations were observed between increasing BPA levels and T2DM in multi-variable adjusted models. Urinary BPA concentrations were higher in older individuals and there was no significant difference between males and females.
[49]	Stahlhut et al., 2018	USA	The administration of a 50 µg/kg/day dose of BPA was linked to an altered glucose-stimulated insulin response. Changes in both the early-phase and late-phase insulin responses were detected in response to BPA.
**Prospective Studies**			
[36]	Sun et al., 2014	USA	BPA levels were associated with incident T2DM in the Nurses’ Health Study (NHS) II but not in the NHS I cohort. BPA exposure may be related to the risk of T2DM in middle-aged but not elderly females. The difference between the studies may be attributed to the different age and menopausal status between cohorts.
[47]	Watkins et al., 2016	Mexico	Urinary BPA is linked to metabolic homeostatic markers during in utero development and peripuberty. In boys, BPA levels were not significantly associated with leptin and C-peptide; however, females showed an 8% increase in leptin levels. Serum glucose was found to be normal in both sexes.
[88]	Bi et al., 2016	China	No significant association between BPA exposure and the risk of incident T2DM was detected. However, a 34-variant genetic risk score significantly modified the effect of BPA exposure on the longitudinal increase in fasting plasma glucose (FPG).
[31]	Shu et al., 2018	China	BPA was positively associated with FPG but not to insulin resistance or ꞵ-cell function. During the follow-up period, baseline BPA levels could not predict the five-year T2D incidence. Furthermore, no significant difference was noted between BPA levels in healthy participants and in subjects diagnosed with T2DM.
[48]	Wang et al., 2019	China	BPA was detected in 89.6% of urine samples with a median concentration of 0.93 ng/L. There was a 3.39% increase in FPG for every 10-fold rise in urinary BPA levels. This also correlated positively with β-cell dysfunction, fasting hyperglycaemia and higher fasting insulin levels in females. The association between BPA and glucose homeostasis markers was not significant in males.
[89]	Rancière et al., 2019	France	Over a nine-year follow-up, a positive association between BPA exposure and incident T2DM was detected independently of traditional risk factors.

**Table 2 ijerph-18-00716-t002:** Summary of the main findings of the meta-analyses investigating the association between BPA and T2DM.

Reference	Author, Year	Population Size	Main Outcome
[37]	Hwang et al., 2018	41,320	This meta-analysis included a total of 16 epidemiological studies (12 cross-sectional, 2 case-control, 1 prospective) that focused on T2DM risk. A positive association between BPA levels and T2DM risk was described, with a pooled OR of 1.28 (95% CI 1.14–1.44).
[13]	Song et al., 2016	18,077	This meta-analysis included a total of 49 epidemiological studies (41 cross-sectional and 8 prospective) that explored the association between multiple endocrine disrupting chemicals (EDCs) and T2DM. Ten studies (8 cross-sectional and 2 prospective cohorts) investigated BPA. Elevated BPA levels were related to increased insulin resistance but not to increased FPG. The relationship between EDCs and T2DM is nonlinear. This meta-analysis concluded that BPA may form part of the environmental factors contributing to developing T2DM.
[89]	Rancière et al., 2015	9291	This meta-analysis included three cross-sectional studies, all reporting urinary BPA levels in quartiles. Pooled ORs (95% CI) for T2DM were 1.33 (1.10–1.61), 1.18 (0.97–1.44) and 1.47 (1.21–1.80) in the second, third, and fourth urinary BPA quartiles respectively, relative to the first quartile.

## Data Availability

Not applicable.

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
