# Peer review of "Bisphenol A and Type 2 Diabetes Mellitus: A Review of Epidemiologic, Functional, and Early Life Factors"

_ijerph, 2021, doi:10.3390/ijerph18020716_

Round 1
Reviewer 1 Report
The manuscript ijerph-1057216 concerns the BPA mechanisms of action in type 2 diabetes mellitus (T2DM) and provides epidemiological data review regarding association between BPA and T2DM risk. The topic is suitable to the journal. The case of BPA clearly shows that chemicals are not being tested enough, prior to use in mass production. Due to the large number of articles focused on BPA-induced adverse health effects that appear every year, there is a need for an up-to-date review of the literature. The manuscript ijerph-1057216 is a well written review on this topic, discussing the most important points and raising the questions to be addressed in the future. Nevertheless, several parts of the manuscript require revision in order to add clarity or to make the manuscript better focused. I have listed some concerns/suggestions below:
- Line 45 – BPA acts via extranuclear oestrogen receptors (ERα and ERβ) in pancreatic islet β-cells, but can also act via membrane GPR30. The involvement of GPR30 in BPA-induced changes in pancreatic beta cells should be included in the manuscript.
- Knowing that single measurement of urinary BPA in most studies reflects recent BPA exposure, what do Authors think that should be done to measure chronic exposure? Is there any available method?
- Additional information regarding comparison between urinary and serum BPA measurements could be useful. What material should be used for T2DM human monitoring studies and exposure evaluation? A short summary of the epidemiological studies review with critical view could be attractive to Readers.
- Did Authors of epidemiological studies pay attention to the elimination of the sources of contamination for BPA as well as other chemicals? This should be included in the manuscript (in additional paragraph?).
- When citing studied, a research model needs to be included (murine/rat/cell line when needed), especially as the authors are aware of the limitations associated with research models used.
- Line 157 – Authors write ‘studies’ however, only 1 citation is provided.
- As authors rightly point that BPA follows non-monotonic dose-response curve, thus in my opinion ‘safe dose of BPA’ is quite unfortunate term. I suggest to withdraw this term from the manuscript.
- Lines 186-187 – please provide citation.
- Lines 186-200- please add additional information regarding the research model and length of exposure, it would be helpful for the Readers to interpret the data
- Gene names should be given in full when appearing in the manuscript-lines 232, 234, 252, 298, 335, 336.
- I suggest to change the order of paragraphs in the manuscript, first describe prenatal exposure, then postnatal (including the study Effect of BPA on glycaemia, insulin resistance and lipids, Effect of BPA on β-cell mass, morphology and function, Effect of BPA on β-cell gene expression, Cellular mechanisms of BPA action).
- Figures should be improved- increasing the font when condensing/reducing the graphic size should allow them to be included in the text not at the end.
- A paragraph regarding mixture effect and BPA substitutes (BPF, BPS, BPAF) should be included.
- A short summary regarding the multitude of BPA mechanisms of action in conclusion section is needed in order to add clarity.
Reviewer 2 Report
Manuscript ID: ijerph-1057216 entitled ‘Bisphenol A and type 2 diabetes mellitus: a review of epidemiologic, functional, and 3 early life factors’ by Farrugia et al provides a comprehensive description of environmental impact of Bisphenol A. However, the junction between mechanisms of BPA actions and the relevance of these mechanisms for clinical studies needs to be refined. Moreover, multiple studies point at existence of different mechanisms of BPA actions that. Should be considered to account for many conflicting results.
- Introduction Line 29:’ relative insulin deficiency’
Since hyperinsulinemia is a prominent feature of type 2 diabetes it would be more correct to talk about ‘functional deficiency of insulin’.
Line 32: ”end organ’. I recommend to avoid this jargon and describe organs. This statement should be supported by references.
Line 34: ‘rare genetic variation’ plural should be used.
Line 43:’ a non-persistent EDC with a short half-life’. The term non-persistent has not been explained and can be simplified by stating its half life. How ling is ‘short’? This info can be important for readers particularly in relation to other sections.
Is BPA binds to ER receptors in the other tissues than pancreas? Is this relevant only for male and postmenopausal women? Please explain a relation with the estrogen: is BPA a competitive activators , partial agonist, or inhibitor?
Adiponectin was not explained and its function was limited to ‘proangiogenic effects on endothelium’.Adiponectin has other functions related to diabetes, were they not affected by BPA?
Introduction required a narrative stating the basis for apparently random selection of information.
3.1.1.’Most of the studies selected are characterized by a cross-sectional study design, although case-control cohort and longitudinal studies were also included’
Since this info is in Table 1 , this table should be organized in a systematic fashion, e.g. by the type of the study designs to allow for comparison.
What study supports statement :’The adverse health consequences of BPA are more pronounced at low 135 levels rather than at higher ones’?
3.2.1.’ Taken together, these findings provide strong support’ . Were the BPA effects on glycemia and insulin secretion tested in ER deficient mice or at least in the presence of ER inhibitors? The described of BPA to estrogen is not causative. Authors should provide a description of studies shoving conclusive evidence of dependence of BPA effects on ER.
Does BPA detoxification in the liver increase inflammation? Increase in hepatic inflammation could be a key factor leading to insulin resistance and pancreatic beta -cell death.
Are glycemic BPA effects sex-specific that will indirectly indicate a relation to ER-dependent mechanism?
From the description of BPA effects with ‘amyloid 215 polypeptide (hIAPP) in a dose-dependent manner’ it is unclear what model was used in these studies/ Was it cell cultures, mouse models? More information is needed to understands the relevance of this study.
‘encouraging satiety’. This statement should be clarified is satiety induced? Is it a function for the brain cells?
If hIAPP is secreted by beta-cells as stated above, why ‘Aggregates of 218 hIAPP can insert into β-cell membrane, causing leakage of cellular contents and eventually apoptosis.’ What is the reason for self-destruction of beta cells ? It obviously cannot take place under physiological conditions, otherwise pancreas will be destroyed within few days. Do these hIAPP aggregates contain BPA, or BPA should be in blood to induce this aggregation? The explanation should be refined and address factors altering physiological response to pathological. Were hIAPP and/or its aggregates measured in clinical studies?
3.2.3. Please provide the levels of BPA found in human blood to understand the physiological relevance of ‘24-hour exposure to 25µg/l BPA results in a downregulation of 225 the pancreatic glucose transporter (SLC2A2) and glucokinase (GCK)’. Of note, GCK was described only later, in 3.2.5.
3.2.4.
The GSIS was not described. This paragraph is also not supported by references for mechanisms of polarization.
The studies in ERb-/- mice suggest that BRA impair insulin secretion. Was low insulin secretion was also found in humans? The discussion should establish the supporting or conflicting relationship between human and animal studies.
3.2.5 First paragraph, please provide a sentence describing study with a ref to one or multiple studies.
‘from patients suffering with T2DM’ The use of ‘suffering with’ appears inappropriate.
‘While oestrogen stimulates glucokinase activity, BPA treatment reduces it under these conditions [58].’
This bring us back to the question are all BPA mechanisms regulating glycemic response depend on ER or other mechanisms are involved? The complexity of these issues needs to be stated in the Introduction and clarified in the manuscript.
‘FASN – the gene encoding fatty 298 acid synthase, a principal regulator of de novo lipogenesis’
FASN is not a regulator, but a key enzyme of de novo lipogenesis. FASN and all other mentioned genes are upregulated by insulin, which is incompatible with the previously proposed decrease of insulin secretion. On the other hands is in line with postnatal studies showing hyperinsulinemia in offspring after BPA exposure. Once more it suggests that BPA acts via different mechanisms.
This paper would benefit from a schematics, suggesting different mechanistic targets of BPA.
3.4
The dose -dependent relationship should be discussed at the beginning of the results section, where it was mentioned but not explained.
3.5
The limitations should also include the possibility of involvement other mechanisms than ER. The endpoints in clinical investigations need to include the markers of other pathways to assess their relevance in humans.
Reviewer 3 Report
In general, define the abbreviations in the first moment they are used and from then on in the writing of the article only use the abbreviation, for example, GDM.
Use italics for in vivo, in vitro, mellitus.
Separate the doses and concentrations of the units (leave a space between the doses or concentrations of the units), for example, 100 µg / Kg / day.
Author Response
Reviewer 3
Comments and Suggestions for Authors
In general, define the abbreviations in the first moment they are used and from then on in the writing of the article only use the abbreviation, for example, GDM. Use italics for in vivo, in vitro, mellitus.
Separate the doses and concentrations of the units (leave a space between the doses or concentrations of the units), for example, 100 µg / Kg / day.
Response: These formatting changes have been implemented.
Sincere thanks for your response and the contribution to our manuscript. We have modified the paper in response to the comments provided to address the points that have been raised.
Round 2
Reviewer 2 Report
I am satisfied with the changes in the revised version